# Incidence and prognostic factors of self-harm and subsequent unnatural death in South Africa: A cohort study

Veronika Whitesell Skrivankova[1], Stephan Rabie[2], Mpho Tlali[3], Naomi Folb[4,5], Chido Chinogurei[3], Sarah Bennett[4], Aimee Wesso[4], Yann Ruffieux[1], Morna Cornell[3], Soraya Seedat[6], Matthias Egger[3,7,8], Mary-Ann Davies[3], Gary Maartens[5], John Joska[2], Andreas D. Haas[1,3]*

1 Institute of Social and Preventive Medicine, University of Bern, Bern, Switzerland, 2 HIV Mental Health Research Unit, Division of Neuropsychiatry, Department of Psychiatry and Mental Health, Neuroscience Institute, University of Cape Town, Cape Town, South Africa, 3 Centre for Integrated Data and Epidemiological Research (CIDER), School of Public Health, University of Cape Town, Cape Town, South Africa, 4 Medscheme, Cape Town, South Africa, 5 Division of Clinical Pharmacology, Department of Medicine, University of Cape Town, Cape Town, South Africa, 6 Department of Psychiatry & South African Research Council Genomics of Brain Disorders Research Unit, Faculty of Medicine and Health Sciences, Stellenbosch University, Cape Town, South Africa, 7 Population Health Sciences, Bristol Medical School, University of Bristol, Bristol, United Kingdom, 8 Department of Infectious Diseases and Hospital Epidemiology, University Hospital Zurich, University of Zurich, Zurich, Switzerland

* andreas.haas@unibe.ch

## Abstract

### Background

Self-harm is a major public health concern globally and in South Africa. Individuals with a history of self-harm are at increased risk of unnatural death, including suicide and fatal accidents. This study investigates the incidence and predictors of self-harm and its role as a predictor for subsequent unnatural death.

### Methods and findings

We conducted a cohort study using insurance claims and vital registration data from beneficiaries of South African medical insurance schemes (2011–2022), aged 10 years or older. We estimated the cause-specific cumulative incidences of healthcare encounters for intentional self-harm (International Classification of Diseases 10th Revision [ICD-10] codes X60-X84) and unnatural deaths (ICD-10 codes V01-Y98), using the Aalen–Johansen method. We assessed predictors of both outcomes using Cox regression. We followed 1,356,119 beneficiaries (median age 33 years, 52.2% female) for a median of 3 years, during which 7,510 (0.6%) had a healthcare encounter for self-harm. The 5-year cumulative incidence of self-harm ranged from 0.2% in males aged 10–14 to 2.1% in females aged 15–24. Sex, age, and mental disorders were strong predictors for self-harm, while HIV was a modest predictor. Among individuals who survived a self-harm event, the

which permits unrestricted use, distribution, and reproduction in any medium, provided the original author and source are credited.

**Data availability statement:** Data were obtained from the International epidemiology Databases to Evaluate AIDS–Southern Africa (IeDEA-SA). Data cannot be made available online because of legal and ethical restrictions. To request data, readers may contact IeDEA-SA for consideration by filling out the online form available at https://www.iedea-sa.org/contact-us/. The code used for the analysis is available on GitHub (https://doi.org/10.5281/zenodo.17119000).

**Funding:** Research reported in this publication was supported by the Swiss National Science Foundation (grant numbers 193381 [AH], and 189498 [ME]; https://www.snf.ch/en) and the US National Institutes of Health (the National Institute of Allergy and Infectious Diseases, the Eunice Kennedy Shriver National Institute of Child Health and Human Development, the National Cancer Institute, the National Institute of Mental Health, the National Institute on Drug Abuse, the National Heart, Lung, and Blood Institute, the National Institute on Alcohol Abuse and Alcoholism, the National Institute of Diabetes and Digestive and Kidney Diseases, and the Fogarty International Center) (grant number U01AI069924 [MD]; https://www.nih.gov). The funders had no role in study design, data collection and analysis, decision to publish, or preparation of the manuscript.

**Competing interests:** The authors declare no conflicts of interest.

**Abbreviations:** aHR, adjusted hazard ratio; CI, confidence intervals; HR, hazard ratio; ICD-10, International Classification of Diseases 10th Revision; IQR, interquartile range; LMICs, low- and middle-income countries; NF-SH, non-fatal self-harm; PMBs, Prescribed Minimum Benefits; STROBE, Strengthening the Reporting of Observational Studies in Epidemiology; WHO, World Health Organization.

five-year cumulative incidence of subsequent unnatural death was 3.43% (95% CI [2.38, 4.76]) for males and 0.77% (95% CI [0.48, 1.19]) for females. Non-fatal self-harm was a strong predictor of subsequent unnatural death in both males (hazard ratio [HR] 7.03, 95% CI [5.27, 9.39]) and females (HR 4.63, 95% CI [3.00, 7.15]). The study's main limitations include potential under-ascertainment of self-harm incidence due to reliance on routine data, and the unavailability of the exact cause of death, preventing analysis of suicide.

## Conclusion

Self-harm is common among beneficiaries of South African private medical insurance, with the highest risk in young females and individuals with mental disorders. These groups may benefit from targeted screening and early intervention. Non-fatal self-harm was a strong predictor of subsequent unnatural death, underscoring the need for suicide-specific brief interventions for individuals presenting with self-harm.

## Author summary

### Why was this study done?

- People who have harmed themselves in the past are more likely to do so again and are at higher risk of dying by suicide or other unnatural means.

- With the right support from healthcare providers, self-harm can often be prevented.

- Although self-harm is common in African countries, there is limited information to guide how health services can best support people affected by it.

### What did the researchers do and find?

- In a cohort of over 1.35 million members of South African medical insurance schemes, each year, around 2 in every 1,000 females and 1 in every 1,000 males accessed healthcare following self-harm.

- The risk of self-harm is especially high among adolescents and young adults aged 15–24, and among people with mental health or substance use problems.

- Individuals with a history of self-harm were at much greater risk of dying from unnatural causes, such as suicide, in the years following the self-harm event. The risk of dying from unnatural causes was particularly high among men.

### What do these findings mean?

- South Africa is facing a high burden of self-harm. Young adults, females, and individuals with mental health or substance use conditions are at increased risk and represent key populations for targeted screening and early intervention.

- Individuals presenting for self-harm, especially males, are at high risk of dying from unnatural causes and may benefit from suicide-specific brief interventions, such as safety planning, motivational techniques, and structured follow-up care.

- The main limitations of the study are that we may have missed some cases of self-harm because we relied on routine medical records, and that we could not examine suicide mortality because the exact cause of death was not available.

## Introduction

Intentional self-harm (hereafter referred to as self-harm) includes a spectrum of behaviors in which individuals deliberately harm themselves. The literature distinguishes three types of self-harm behaviors: *non-suicidal self-injury*, defined as deliberate, directly injurious behavior (e.g., cutting, or burning) without any intent to die; *suicide attempts*, defined as non-fatal potentially self-injurious behaviors with at least some intent to die; and *suicide*, defined as fatal self-directed injurious behavior with intent to die [1,2].

Self-harm contributes substantially to the global disease burden and places considerable strain on health systems, especially in low- and middle-income countries (LMICs) like South Africa, where self-harm rates are high [3,4] and resources for prevention and care are scarce [5,6]. The crude suicide rate in South Africa in 2021 was estimated at 35.4 per 100,000 population among males, 9.9 among females, placing the country among those with the highest rates globally [7]. Lifetime prevalence of suicide attempts in South Africa is 3.8% among females and 1.8% among males [8]. Non-suicidal self-injury, which predominantly affects adolescents and young adults [1,2], is also highly prevalent with lifetime estimates ranging from 33% to 56% [9,10].

Self-harm cases presenting to emergency departments in South Africa are predominantly episodes of self-poisoning without evidence of suicidal intent or suicide attempts. At hospitals in Cape Town and KwaZulu-Natal, poisoning, mainly with medication, accounted for 72%–80% of presentations, while highly lethal methods such as hanging or laceration were less common (6%–14%). Across all methods, most patients (66%–78%) reported no intent to die [11,12]. While non-suicidal self-injury is common among adolescents in South Africa, it is infrequently seen in hospital or emergency settings [11,12].

Self-harm behaviors are often recurrent and may escalate in severity [2,13]. Individuals with a history of self-harm are at increased risk of unnatural death including suicide and accidental death [14–17]. Non-suicidal self-injury has consistently been shown to predict suicidal ideation and suicide attempts [18]. Intentional self-poisoning, regardless of intent, is a strong predictor subsequent suicide [14]. The risk of recurrent suicidal behavior after a suicide attempt is well documented, with 16% reattempting and 3% dying by suicide within one year [13,15]. Beyond suicide, individuals with a self-harm history also face increased risks of accidental, alcohol-related, and drug-related mortality [16,17].

The World Health Organization (WHO) recommends early identification of individuals at risk of self-harm and suicide, and management, support, and monitoring of those with a history of self-harm as key effective suicide prevention strategies [19]. The organization urges countries to implement evidence-based strategies tailored to local contexts and informed by local data [19], to achieve the international target of reducing suicide mortality by one third from 2015 levels by 2030 [19,20]. However, current data on self-harm in South Africa are scarce, primarily derived from small, cross-sectional, single-center studies without control groups [11,12,21–23]. The existing studies have not estimated the incidence or predictors of self-harm and subsequent mortality. Such estimates are essential to support early identification of at-risk groups and to inform the development of evidence-based strategies for managing individuals presenting with self-harm, tailored to their risk of subsequent unnatural death.

To address these gaps, we analyzed longitudinal data from over 1 million beneficiaries of South African medical insurance schemes. We aimed to estimate self-harm incidence, identify its predictors, and evaluate non-fatal self-harm as

a predictor of subsequent unnatural death. Our analyses were not designed to identify causal (etiologic) risk factors. Instead, we aimed to identify individual-level characteristics that serve as risk markers, indicating increased future self-harm or mortality risk, without necessarily implying causality [24].

## Methods

### Study design and participants

We conducted a cohort study using longitudinal reimbursement claims data from a South African medical insurance scheme (hereafter referred to as the medical scheme) and a private sector HIV disease management program (hereafter referred to as the HIV program). The design and objectives of this study align with prognostic factor research [24], as defined in the PROGnosis RESearch Strategy (PROGRESS) framework [25]. We refer to prognostic factors as 'predictors' throughout the manuscript, as study a general population of medical insurance beneficiaries, not individuals with a specific disease. We conducted two primary analyses: (1) the *self-harm analysis*, investigating the incidence and predictors of intentional self-harm, and (2) the *mortality analysis*, studying non-fatal self-harm as a predictor of subsequent unnatural death. We included individuals aged 10 years or older who were covered by the medical scheme (1 Jan 2011–30 Jun 2020) or enrolled in the HIV program (1 Jan 2011–1 Oct 2022) (Fig 1). We excluded HIV program members also included in the medical scheme cohort, and individuals with unknown sex. From mortality analysis, we excluded individuals not linkable to the vital registration system. We conducted analyses according to a pre-specified protocol (S1 Appendix), and list protocol deviations in S2 Appendix. This study is reported as per the Strengthening the Reporting of Observational Studies in Epidemiology (STROBE) guideline (S1 STROBE Checklist). The Human Research Ethics Committee of

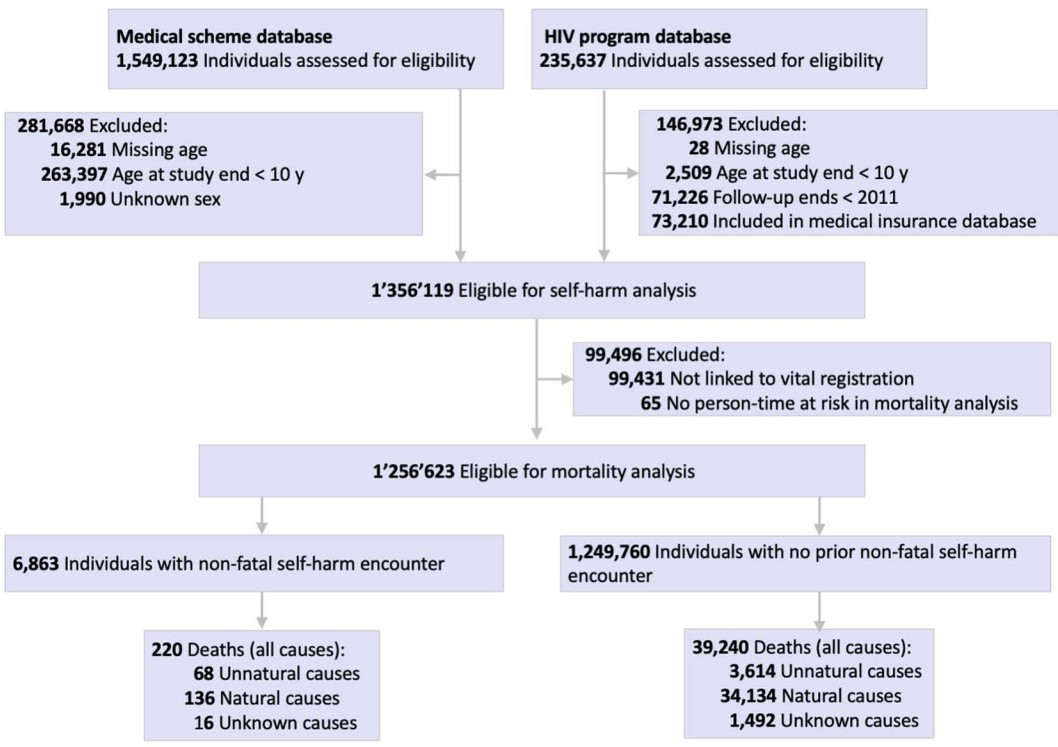

**Fig 1. Study inclusion flowchart.**

the University of Cape Town (HREC REF Number: 84/2006) and the Cantonal Ethics Committee of the Canton of Bern granted (KEK-Nr: 150/14) permission to analyze data.

## Data sources

We combined data from the medical scheme and HIV program to increase sample size, given the rarity of unnatural death after self-harm. The dataset included sociodemographic characteristics, reimbursement claims for inpatient and outpatient care with International Classification of Diseases 10th Revision (ICD-10) diagnoses, pharmacy claims, and laboratory results. These data were deterministically linked to vital registration records (1 Jan 2011–26 Jan 2021) from the National Population Register using national identifiers. Death registration is estimated to be 93% complete among adults and slightly lower among children [26,27]. Vital registration data included date of death, and cause of death (natural, unnatural, or unknown).

## Setting

The medical scheme is open to all South African residents but requires paid membership. Most residents do not have private insurance and rely on public sector services [28]. The HIV program includes individuals with HIV whose medical schemes have contracted the program to manage their HIV care. Medical scheme beneficiaries generally have higher socio-economic status and may face fewer psychosocial stressors of self-harm than individuals without insurance [29]. In addition to medical scheme benefits, all beneficiaries are entitled to Prescribed Minimum Benefits (PMBs), which include essential services. For self-harm, PMBs cover up to 3 days of inpatient care or six outpatient visits per episode, including treatment of injuries, complications, and underlying conditions [30].

## Outcomes

In the self-harm analysis, the outcome was an individual's first observed healthcare encounter for intentional self-harm (ICD-10 codes X60-X84) [31]. Repeated self-harm events were not considered. In the mortality analysis, the outcome was death from unnatural causes (ICD-10 codes V01-Y98), including suicides and fatal accidents.

## Predictors

We defined predictors as individual-level characteristics, measured before the outcome among people with a given start point, that indicate increased risk of future self-harm or mortality, without necessarily implying causality [24,32]. In the self-harm analysis, we included sex, age group (10–14, 15–24, 25–39, 40+ years), HIV status, and mental disorder diagnoses as potential predictors. We defined HIV status using antiretroviral therapy codes (excluding prophylaxis), HIV-related ICD-10 diagnoses (B20–24, F02.4, R75, Z21), relevant lab tests (e.g., viral load, CD4), or enrollment in the HIV program. Mental disorders were grouped by ICD-10 codes into organic mental disorders (F00–09), substance use disorders (F10–16, F18–19), psychotic disorders (F20–29), bipolar disorder (F31), depression (F32–34.1), anxiety (F40–48), personality disorders (F60–69), and other mental disorders (F50–59, F70–99). For individuals with both depression and bipolar disorder, only the bipolar diagnosis was considered from the time of diagnosis onward.

In the mortality analysis, we considered a healthcare encounter for non-fatal self-harm as the main predictor. As a secondary predictor, we classified self-harm encounters by lethality of the self-harm method, based on evidence that individuals who use highly lethal methods at their index event are at greatest risk of suicide [33]. High lethality methods included hanging (X70), gas poisoning (X67), jumping (X80–X81), firearms (X72–X75), and drowning (X71). Less lethal methods included drug or chemical poisoning (X60–X66, X68–X69), fire-related injuries (X76–X77), motor vehicle accidents (X82), sharp or blunt object injuries (X78–X79), and other or unspecified methods (X83–X84) [34].

## Statistical analysis

Baseline was defined as the latest of medical scheme or HIV program enrollment, the individual's 10th birthday, or January 1, 2011. Age group, non-fatal self-harm, HIV status, and mental disorders were modeled as time-varying covariates, with individuals considered exposed from the first recorded occurrence. Data preparation and statistical analyses were done using R version 4.3.1 and Stata version 18.

**Self-harm analysis.** Individuals were followed from baseline until the earliest of the end of insurance coverage, medical scheme database closure, death, or first observed self-harm encounter. We calculated crude incidence rates per 100,000 person-years by sex and age group with 95% confidence intervals (CIs) based on the Poisson distribution. Using Royston-Parmar flexible parametric survival models [35], we estimated self-harm incidence as a continuous function of age, using a spline with 5 degrees of freedom, stratified by sex, and HIV status. We estimated cause-specific cumulative incidence of self-harm using the Aalen–Johansen estimator, treating death as a competing event and stratifying by age group and sex [36]. Individuals exited the risk set upon first self-harm. We used Cox regression to estimate age- and sex-adjusted and fully adjusted hazard ratios (HRs) for predictors of self-harm, treating death as a censoring event. Fully adjusted models included age, sex, HIV, and mental disorders.

**Mortality analysis.** Individuals were followed from baseline until the earliest of the end of insurance coverage, medical scheme database closure, vital registration database closure, or death. We estimated the cumulative incidence of unnatural death using the Aalen–Johansen estimator, stratified by sex and self-harm history, with natural and unknown causes treated as competing events. Using Cox models, we estimated cause-specific age-adjusted and fully adjusted HRs to evaluate non-fatal self-harm as a predictor for subsequent unnatural death, treating natural or unknown causes as censoring events. Age-adjusted models used age as the underlying time scale and were stratified by sex, with separate models fitted for males and females. Fully adjusted models additionally controlled for mental disorders.

**Post-hoc analyses.** We conducted the following additional post-hoc analyses: First, we assessed whether the lethality of the method used at the index self-harm event predicted subsequent unnatural death. Second, we used flexible parametric survival models to estimate age-adjusted, time-varying HRs for non-fatal self-harm. Third, we fitted a combined Cox model including both males and females, using age as the time scale and adjusting for sex to estimate association between prior non-fatal self-harm and unnatural death across sexes. In addition, we fitted the same model with an interaction term between sex and prior non-fatal self-harm to formally test for effect modification. Finally, we used descriptive statistics to compare healthcare utilization prior to self-harm encounters with that of matched controls prior to hypothetical encounter dates. For this analysis, each self-harm case was matched to two randomly selected controls who had no self-harm encounter. Matching criteria included the same baseline year, age group (10–14, 15–19, 20–39, 40–59, 60+), and duration of follow-up (<2 years, 2–4 years, 4–6 years, 6–8 years, or 8+ years). The hypothetical event date for controls was set as their baseline date plus the interval from baseline to the self-harm encounter of the matched case. We calculated standardized differences to assess group differences, interpreting values <0.1 as not meaningful, >0.2 as moderate, and >0.5 as large [37]. Further details are provided in Text A in S3 Appendix.

**Sensitivity analyses.** In post-hoc sensitivity analyses, we repeated all primary analyses in the medical scheme cohort only, to assess the impact of oversampling individuals with HIV due to the inclusion of HIV program data. To assess generalizability to individuals not linked to the vital registration system, we compared those included versus excluded from the mortality analysis. Self-harm incidence was compared using Cox regression adjusted for age, sex, HIV status, and mental disorders.

## Results

### Characteristics of the study population

We followed 1,356,119 individuals (Fig 1) (709,073 females, 647,046 males) for a median of 3.3 years (interquartile range [IQR] [1.2, 6.8]). Median age at baseline was 33 years (IQR 21–46). By the end of follow-up, 177,193 (13.1%)

had HIV diagnosis and 348,157 (25.7%) had a mental health diagnosis, most commonly anxiety (16.1%) or depression (13.5%) (Table 1).

## Characteristics of individuals with self-harm

Among 7,510 individuals (0.6%) with a self-harm encounter (5,208 females, 2,302 males), most were hospital admissions (80%). Less lethal methods, mainly poisoning, accounted for 97.8% of cases; highly lethal methods were rare (2.2%). Median age at self-harm encounter was 31 years (IQR [20, 40]). Half of the cases (50.4%) had received a mental health diagnosis before the self-harm encounter, with another 23.7% being diagnosed within 7 days after the encounter (Table 2).

**Table 1. Characteristics of the study population by sex.**

| | Female | Male | Total |
|---|---|---|---|
| | *N*=709,073 (52.2%) | *N*=647,046 (47.8%) | *N*=1,356,119 (100.0%) |
| **Characteristics at baseline** | | | |
| Age, years | | | |
| 10–14 | 104,952 (14.8%) | 107,055 (16.5%) | 212,007 (15.6%) |
| 15–24 | 111,327 (15.7%) | 95,318 (14.7%) | 206,645 (15.2%) |
| 25–39 | 237,889 (33.5%) | 204,255 (31.6%) | 442,144 (32.6%) |
| 40+ | 254,905 (35.9%) | 240,418 (37.2%) | 495,323 (36.5%) |
| Median (IQR) | 33 (21, 46) | 33 (20, 46) | 33 (21, 46) |
| **Characteristics at the end of follow-up** | | | |
| HIV status | | | |
| HIV–negative | 606,426 (85.5%) | 572,500 (88.5%) | 1,178,926 (86.9%) |
| HIV–positive | 102,647 (14.5%) | 74,546 (11.5%) | 177,193 (13.1%) |
| Any mental health disorders | 206,878 (29.2%) | 141,279 (21.8%) | 348,157 (25.7%) |
| Organic mental disorder | 5,972 (0.8%) | 4,863 (0.8%) | 10,835 (0.8%) |
| Substance use disorder | 2,663 (0.4%) | 6,540 (1.0%) | 9,203 (0.7%) |
| Psychotic disorder | 2,795 (0.4%) | 2,687 (0.4%) | 5,482 (0.4%) |
| Bipolar disorder | 13,895 (2.0%) | 7,933 (1.2%) | 21,828 (1.6%) |
| Depression | 116,173 (16.4%) | 66,538 (10.3%) | 182,711 (13.5%) |
| Anxiety disorder | 135,820 (19.2%) | 82,761 (12.8%) | 218,581 (16.1%) |
| Personality disorder | 920 (0.1%) | 804 (0.1%) | 1,724 (0.1%) |
| Other mental health disorders | 33,601 (4.7%) | 34,136 (5.3%) | 67,737 (5.0%) |
| Self-harm encounters | | | |
| Event of undetermined intent | 5,255 (0.7%) | 7,777 (1.2%) | 13,032 (1.0%) |
| Intentional self-harm | 5,208 (0.7%) | 2,302 (0.4%) | 7,510 (0.6%) |
| Follow-up time, years | | | |
| Median (IRQ) | 3.4 (1.2,7.0) | 3.3 (1.2,6.7) | 3.3 (1.2,6.8) |
| Linked to National Population Register | 651,866 (91.9%) | 604,822 (93.5%) | 1,256,688 (92.7%) |
| Died during follow-up* | 17,495 (2.7%) | 22,034 (3.6%) | 39,529 (3.1%) |
| Natural cause* | 15,921 (2.4%) | 18,349 (3.0%) | 34,270 (2.7%) |
| Unnatural cause* | 923 (0.1%) | 2,759 (0.5%) | 3,682 (0.3%) |
| Unknown cause* | 651 (0.1%) | 926 (0.2%) | 1,577 (0.1%) |

Data are *n* (%) unless otherwise stated.

*Among those linked to National Population Register.

Abbreviation: IQR, interquartile range.

**Table 2. Characteristics of individuals with healthcare encounter for self-harm.**

| | Female | Male | Total |
|---|---|---|---|
| | *N*=5,208 (69.3%) | *N*=2,302 (30.7%) | *N*=7,510 (100.0%) |
| Self-harm encounter type | | | |
| Hospital admission | 4,313 (82.8%) | 1,699 (73.8%) | 6,012 (80.1%) |
| Outpatient encounter | 895 (17.2%) | 603 (26.2%) | 1,498 (19.9%) |
| Self-harm method | | | |
| Highly lethal | 57 (1.1%) | 105 (4.6%) | 162 (2.2%) |
| Self-poisoning by exposure to gasses | 5 (0.1%) | 22 (1.0%) | 27 (0.4%) |
| Self-harm by hanging strangulation and suffocation | 16 (0.3%) | 30 (1.3%) | 46 (0.6%) |
| Self-harm by drowning and submersion | 7 (0.1%) | 1 (0.0%) | 8 (0.1%) |
| Self-harm by firearms or explosives | 17 (0.3%) | 38 (1.7%) | 55 (0.7%) |
| Self-harm by jump. from heights, or before moving objects | 12 (0.2%) | 14 (0.6%) | 26 (0.3%) |
| Less lethal | 5,151 (98.9%) | 2,197 (95.4%) | 7,348 (97.8%) |
| Self-poisoning by an exposure to substances excl. gasses | 4,901 (94.1%) | 1,940 (84.3%) | 6,841 (91.1%) |
| Nonopioid analgesics, antipyretics and antirheumatics | 1,062 (21.9%) | 296 (15.4%) | 1,358 (20.0%) |
| Antiepileptic, sedative-hypnotic, antiparkinsonism and other psychotropic drugs | 1,098 (22.6%) | 379 (19.7%) | 1,477 (21.8%) |
| Narcotics and psychodysleptics | 133 (2.7%) | 90 (4.7%) | 223 (3.3%) |
| Other drugs acting on the autonomic nervous system | 139 (2.9%) | 55 (2.9%) | 194 (2.9%) |
| Other and unspecified drugs, medicaments and biological substances | 1,988 (40.9%) | 765 (39.7%) | 2,753 (40.6%) |
| Alcohol | 88 (1.8%) | 102 (5.3%) | 190 (2.8%) |
| Organic solvents and halogenated hydrocarbons | 65 (1.3%) | 38 (2.0%) | 103 (1.5%) |
| Pesticides | 107 (2.2%) | 94 (4.9%) | 201 (3.0%) |
| Other and unspecified chemicals and noxious substances | 178 (3.7%) | 107 (5.6%) | 285 (4.2%) |
| Self-harm by steam, hot vapors, and hot objects | 5 (0.1%) | 3 (0.1%) | 8 (0.1%) |
| Self-harm by sharp object | 98 (1.9%) | 92 (4.0%) | 190 (2.5%) |
| Self-harm by blunt object | 28 (0.5%) | 57 (2.5%) | 85 (1.1%) |
| Self-harm by smoke, fire and flames | 17 (0.3%) | 11 (0.5%) | 28 (0.4%) |
| Self-harm by crashing of motor vehicle | 3 (0.1%) | 5 (0.2%) | 8 (0.1%) |
| Self-harm by other or unspecified means | 99 (1.9%) | 89 (3.9%) | 188 (2.5%) |
| Time from baseline to first self-harm encounter, years | | | |
| Median (IRQ) | 2.6 (1.1, 4.8) | 2.7 (1.2, 4.8) | 2.6 (1.1, 4.8) |
| Age at self-harm event, years | | | |
| 10–14 | 259 (5.0%) | 83 (3.6%) | 342 (4.6%) |
| 15–24 | 1,658 (31.8%) | 629 (27.3%) | 2,287 (30.5%) |
| 25–39 | 2,096 (40.2%) | 872 (37.9%) | 2,968 (39.5%) |
| 40+ | 1,195 (22.9%) | 718 (31.2%) | 1,913 (25.5%) |
| Median (IQR) | 30 (20, 39) | 33 (21, 43) | 31 (20, 40) |
| Mental health diagnoses before the self-harm encounter* | | | |
| Any mental health disorders | 2,652 (50.9%) | 1,130 (49.1%) | 3,782 (50.4%) |
| Organic mental disorder | 57 (1.1%) | 43 (1.9%) | 100 (1.3%) |
| Substance use disorder | 94 (1.8%) | 180 (7.8%) | 274 (3.6%) |
| Psychotic disorder | 66 (1.3%) | 40 (1.7%) | 106 (1.4%) |
| Bipolar disorder | 501 (9.6%) | 207 (9.0%) | 708 (9.4%) |
| Depression | 1,944 (37.3%) | 746 (32.4%) | 2,690 (35.8%) |
| Anxiety disorder | 1,692 (32.5%) | 636 (27.6%) | 2,328 (31.0%) |
| Personality disorder | 17 (0.3%) | 12 (0.5%) | 29 (0.4%) |
| Other mental health disorders | 451 (8.7%) | 235 (10.2%) | 686 (9.1%) |

*(Continued)*

**Table 2.** (Continued)

| | Female N=5,208 (69.3%) | Male N=2,302 (30.7%) | Total N=7,510 (100.0%) |
|---|---|---|---|
| New mental health diagnoses within 7 days of self-harm encounter** | | | |
| Any mental health disorders | 1,297 (24.9%) | 485 (21.1%) | 1,782 (23.7%) |
| Organic mental disorder | 35 (0.7%) | 27 (1.2%) | 62 (0.8%) |
| Substance use disorder | 67 (1.3%) | 95 (4.1%) | 162 (2.2%) |
| Psychotic disorder | 25 (0.5%) | 26 (1.1%) | 51 (0.7%) |
| Bipolar disorder | 176 (3.4%) | 75 (3.3%) | 251 (3.3%) |
| Depression | 1,461 (28.1%) | 521 (22.6%) | 1,982 (26.4%) |
| Anxiety disorder | 321 (6.2%) | 124 (5.4%) | 445 (5.9%) |
| Personality disorder | 11 (0.2%) | 9 (0.4%) | 20 (0.3%) |
| Other mental health disorders | 25 (0.5%) | 15 (0.7%) | 40 (0.5%) |

Data are n (%) unless otherwise stated.

*Assessed on the day before the self-harm encounter,

**New mental health diagnoses were defined as those received within 7 days after the self-harm encounter, with no prior diagnosis in that category.

Abbreviations: IQR, interquartile range.

## Self-harm incidence

The incidence of self-harm encounters per 100,000 person-years was 183 (95% CI [178, 188]) in females, and 90 (95% CI [87, 94]) in males, with variation by age groups (Table A in S3 Appendix). Rates peaked at age 19 in females (522, 95% CI [493, 553]) and at 21 in males (208, 95% CI [191, 226]), then declined (Fig 2A). After 5 years, the cumulative incidence of self-harm ranged from 0.2% in males 10–14 years to 2.1% in females aged 15–24 (Fig 2B and Table B in S3 Appendix). Incidence rates were slightly higher among people with HIV under 40 years of age compared to their counterparts without HIV (Fig A in S3 Appendix).

## Predictors of self-harm

In fully adjusted models, younger age and female sex were strong predictors of self-harm: compared to those aged ≥40, individuals aged 15–24 had over 5-fold higher risk (adjusted hazard ratio [aHR] 5.43, 95% CI [5.10, 5.78]), and those aged 25–39 had 2.6-fold higher risk (aHR 2.57, 95% CI [2.42, 2.72]); females had nearly twice the risk of males (aHR 1.79, 95% CI [1.71, 1.88]). HIV status was a modest predictor (aHR 1.19, 95% CI [1.12, 1.26]). All mental disorders were predictors of self-harm, with considerable variation in effect size, with aHRs ranged from 1.20 (95% CI [0.98, 1.47]) for psychotic disorders to 8.70 (95% CI [7.97, 9.52]) for bipolar disorder. Adjusted HRs were higher across all disorders in age- and sex-adjusted models (Fig 3).

## Health care utilization before self-harm

Two-thirds of beneficiaries (69.9%, 5,158/7,379) visited an inpatient or outpatient setting for any reason within 30 days before the self-harm event, including 19.3% who were seen for a mental health condition, and 21.6% who claimed psychiatric medication. All estimates were higher among individuals with self-harm encounter compared to matched controls (Table C in S3 Appendix).

## Mortality after non-fatal self-harm

The mortality analysis included 1,256,623 individuals, excluding 99,431 who were not linked to the vital registration system and 65 who had no person-time at risk in the mortality analysis. Among the included individuals, 6,863 had a non-fatal

**A**

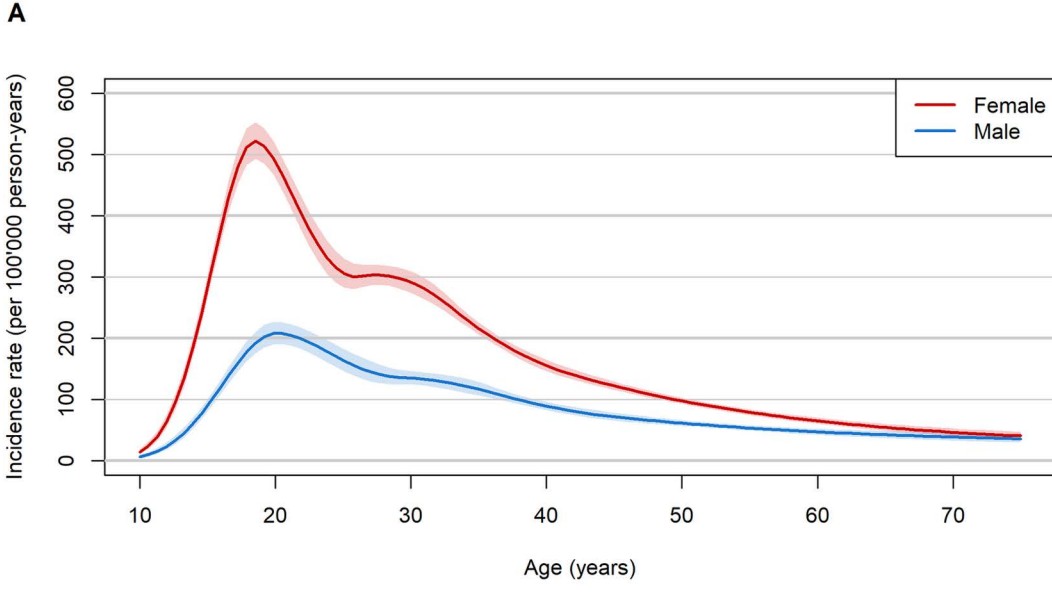

**B**

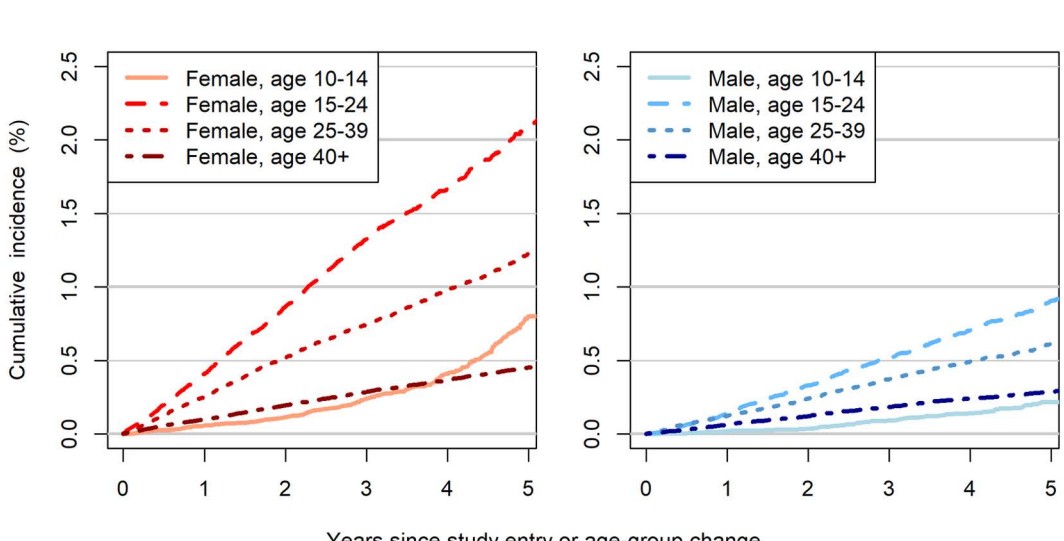

**Fig 2. Incidence of self-harm by age and sex.** Panel **A** presents the incidence per 100,000 person-years as a continuous function of age, for females and males. Panel **B** shows the Aalen–Johansen estimate of cause-specific cumulative incidence over time since baseline or age-group change, stratified by sex and age group. Individuals were censored at the time of their first documented self-harm event. Repeated self-harm events were not considered.

self-harm encounter (4,756 [69.3%] female and 2,107 [30.7%] male), and 1,249,760 had no prior non-fatal self-harm encounter. In the group with a non-fatal self-harm encounter, 68 subsequent unnatural deaths occurred during follow-up (Fig 1). Two years after a non-fatal self-harm encounter, the cumulative incidence of unnatural death was 0.35% (95% CI [0.20, 0.60]) in females and 1.90% (95% CI [1.32, 2.63]) in males, compared with 0.06% (95% CI [0.05, 0.06]) in females and 0.21% (95% CI [0.20, 0.23]) in males without prior self-harm (Fig 4A and Table D in S3 Appendix). The cumulative

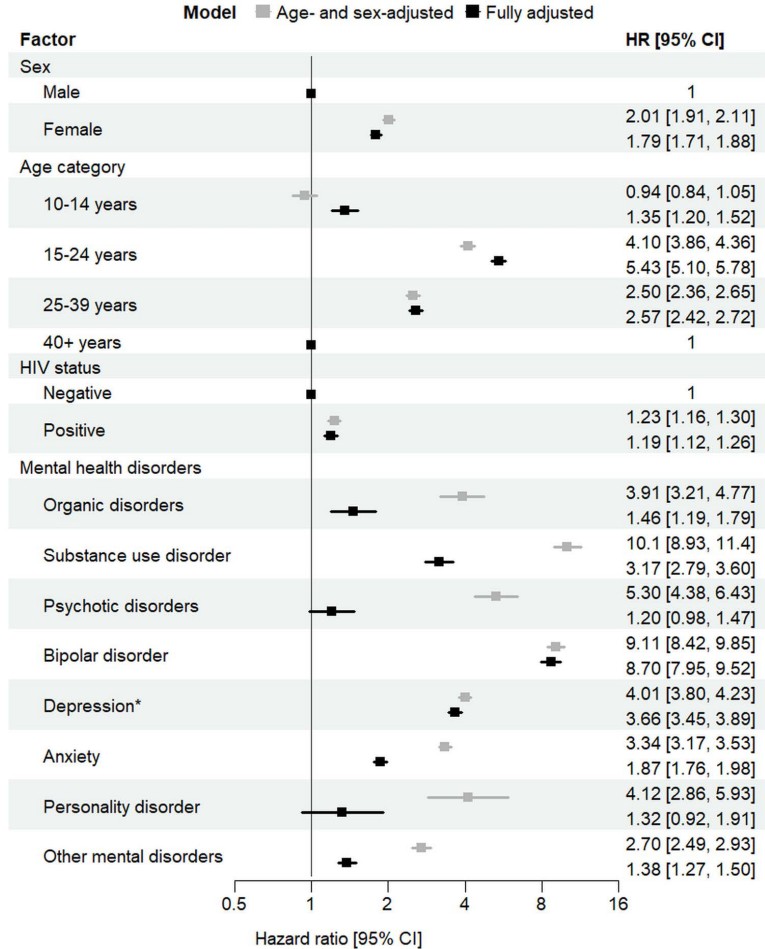

**Fig 3. Predictors of self-harm.** The figure shows age- and sex-adjusted and fully adjusted hazard ratios for predictors of self-harm. Individuals were censored at the time of their first documented self-harm event. Repeated self-harm events were not considered. The error bars represent 95% confidence intervals. Fully adjusted models controlled for age group, sex, HIV, and all mental disorders. * For individuals diagnosed with both depression and bipolar disorder, only the bipolar disorder diagnosis was considered from the date of bipolar diagnosis onwards. Abbreviations: CI, confidence intervals; HR, hazard ratio.

incidence of unnatural death within two years after a non-fatal self-harm encounter reached 3.84% (95% CI [1.01, 9.89]) in males and 4.26% (95% CI [0.76, 13.00]) in females among those who used highly lethal methods (Fig 4B and Table D in S3 Appendix).

### Non-fatal self-harm as predictor for unnatural death

Health care encounters for non-fatal self-harm were strongly predictive of subsequent mortality from unnatural causes in both sexes. Among females, the risk of unnatural death following a non-fatal self-harm encounter was almost five times higher than in females of the same age without a prior non-fatal self-harm encounter (HR 4.63, 95% CI [3.00, 7.15]; Fig 5A). Among males, the risk was seven times higher (HR 7.03, 95% CI [5.27, 9.39]; Fig 5A). In both sexes, HRs were attenuated in fully adjusted models (Fig 5A). Time-varying HRs for non-fatal self-harm as a predictor of unnatural death are shown in Fig B in S3 Appendix. In a combined model including both sexes and adjusting for age and sex, non-fatal

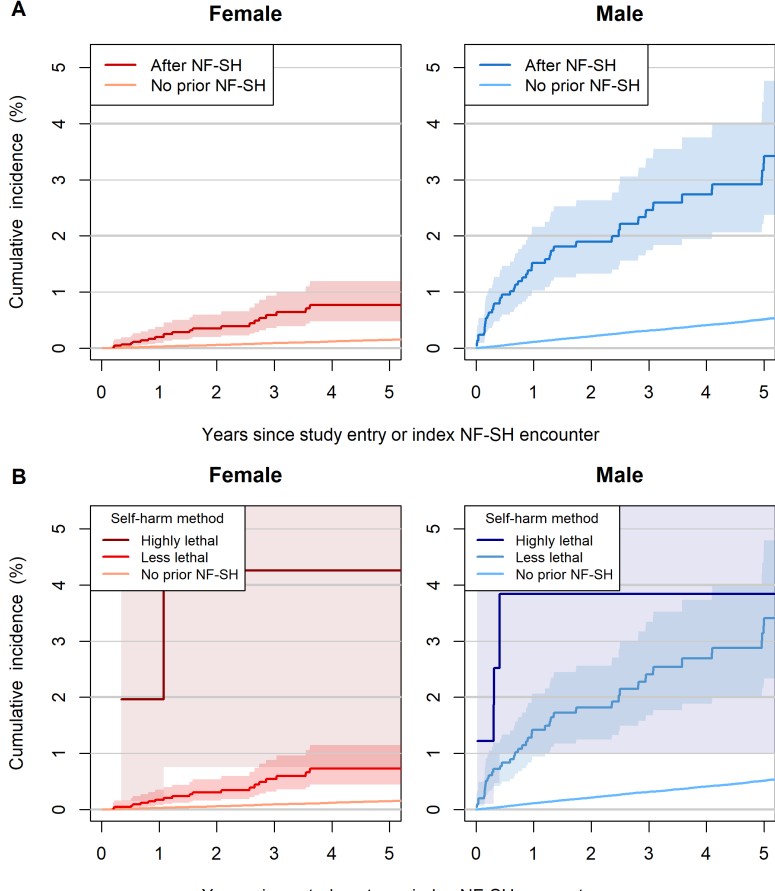

**Fig 4. Cumulative incidence of unnatural death by sex and self-harm history.** Aalen–Johansen estimate with 95% confidence intervals for cause-specific cumulative incidence of unnatural death over time since baseline or index non-fatal intentional self-harm (NF-SH) encounter, stratified by sex and **(A)** NF-SH, **(B)** self-harm method at index encounter. Abbreviations: NF-SH, non-fatal self-harm.

self-harm was associated with a 6-fold higher risk of unnatural death (HR 6.01, 95% CI [4.73, 7.65]). The interaction between sex and prior self-harm showed weak statistical evidence for effect modification ($p = 0.079$).

## Self-harm method as predictor for unnatural death

Use of highly lethal methods at the index self-harm encounter was a very strong predictor of subsequent unnatural death. Compared with individuals of the same sex and age without prior non-fatal self-harm, the age-adjusted HR was 29.1 (95% CI [7.27, 117.0]) for females and 11.20 (95% CI [3.59, 34.6]) for males (Fig 5B). In both sexes, HRs were attenuated in fully adjusted models (Fig 5B).

## Sensitivity analyses

Compared with the medical scheme cohort, the HIV program cohort included a higher proportion of females and fewer individuals younger than 25 years (Table E in S3 Appendix). Main findings remained consistent in sensitivity analyses restricted to the medical scheme cohort, although the strength of associations varied somewhat (Text B and Figs C–F in S3 Appendix). One notable divergence was observed in the analysis of self-harm method as a

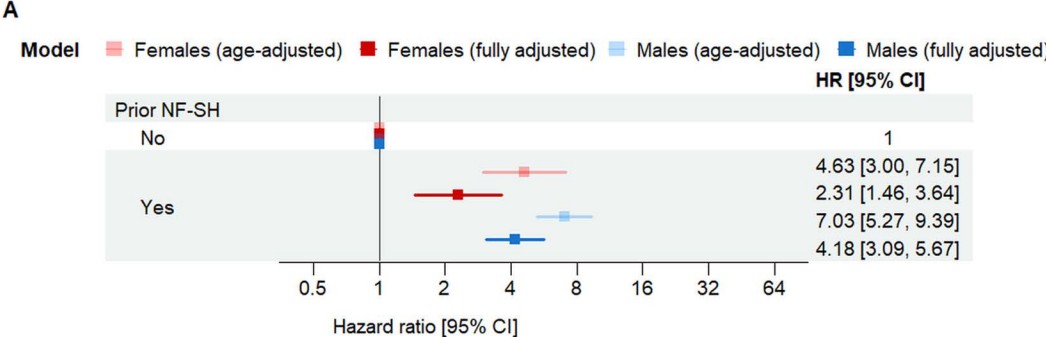

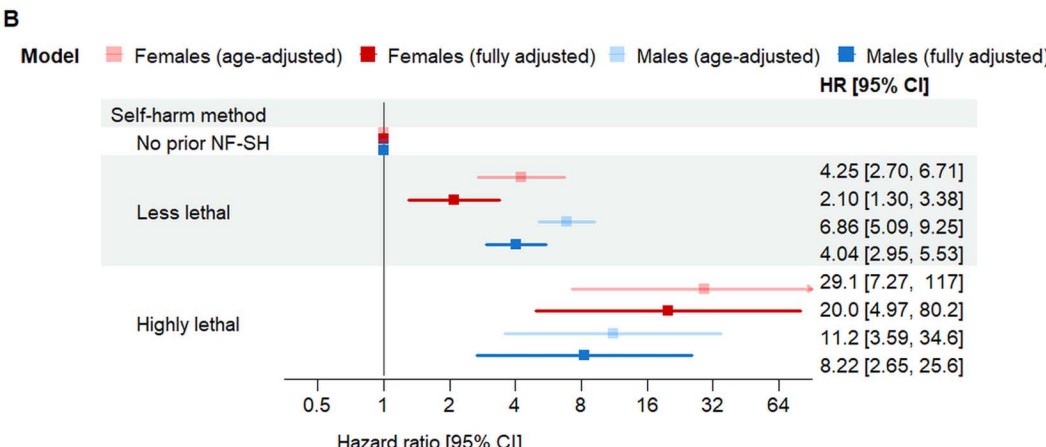

**Fig 5. Hazard ratios for unnatural death comparing individuals with and without a history of non-fatal self-harm by sex.** All models were fitted separately for males and females. Age-adjusted estimates control for age using it as the timeline in Cox model. Fully adjusted models additionally control for HIV status, mental health diagnoses. Error bars represent 95% confidence intervals. Panel **A** shows hazard ratios for a previous non-fatal intentional self-harm (NF-SH) encounter. Panel **B** shows hazard ratios by method of self-harm (highly lethal vs. less lethal) used during the index self-harm event. Individuals with no prior NF-SH encounter serve as the reference group in both analyses. Abbreviations: NF-SH, non-fatal self-harm; CI, confidence intervals; HR, hazard ratio.

predictor of unnatural death, where findings were not robust in sensitivity analyses: associations for highly lethal methods were stronger in females but attenuated and not statistically significant in males (Text B and Fig G in S3 Appendix).

Individuals excluded from mortality analysis were younger, had slightly higher HIV prevalence, lower prevalence of depression and anxiety (Text C and Table F in S3 Appendix), and a higher incidence of self-harm (aHR 1.25, 95% CI [1.15, 1.36]) than those included.

## Discussion

In this analysis of South African medical insurance data, self-harm incidence was high, predominantly involving self-poisoning, with few cases using highly lethal methods. Key predictors of self-harm included young age, female sex, and mental health diagnoses. Non-fatal self-harm was a strong predictor of subsequent unnatural death in both sexes. As the analyses were prognostic in nature, the identified predictors should not be interpreted as causal but as risk markers to identify individuals at elevated risk.

Self-harm incidence rates in this study (females: 183; males: 90 per 100,000 person-years) were twice as high as 2021 estimates for LMICs (females: 91; males: 42 per 100,000), aligning more closely with rates reported in high-income countries (females: 143; males: 112 per 100,000) [3,38]. Consistent with other studies [39,40], incidence peaked among females aged 15–24. Our findings also align with limited South African research suggesting a higher burden of non-fatal self-harm among females [41], underscoring the substantial risk in young women.

Mental disorders were strong predictors of self-harm. Half of the self-harm cases in our study had a pre-existing mental health diagnosis, and 24% were newly diagnosed within 7 days post-incident, suggesting previously unrecognized illness. The psychiatric morbidity observed in this study is slightly higher than in other LMICs, where fewer than half of those engaging in suicidal behavior had a psychiatric disorder [42]. In contrast, in Western countries, 90% of hospital-presenting self-harm cases had a mental disorder [43]. Growing evidence from LMICs highlights the role of socioeconomic and psychosocial stressors in self-harm [21,22,44], indicating that prevention strategies must address both mental illness and psychosocial factors [40,45,46].

Over two-thirds of individuals who engaged in self-harm had outpatient care contact in the 30 days preceding the episode, and one-fifth received mental healthcare during this period. These findings align with South African data showing that many individuals seek care shortly before suicide [47], underscoring outpatient care as a key setting for prevention. Nurses, counselors, and social workers are well placed to screen for suicidality but require training and supervision to intervene effectively without overburdening the system [48]. Estimates from our study on predictors of self-harm could help inform targeted screening approaches.

Non-fatal self-harm was a strong predictor of subsequent unnatural death in both sexes. Due to non-proportional hazards, the HRs for non-fatal self-harm should be interpreted as weighted averages of time-varying effects over the follow-up period [45]. We present two adjusted analyses for this predictor which should be interpreted differently based on their adjustment sets. The sex-stratified, age-adjusted analysis indicates the excess risk of unnatural death among individuals with a history of self-harm compared to general population controls of the same age and sex, without adjusting for factors such as mental disorders that form the prognostic profile of people who engage in self-harm. We consider this analysis the most informative, as it captures the overall excess mortality burden associated with self-harm, reflecting the real-world distribution of suicide risk factors faced by affected individuals. This includes a high prevalence of psychiatric illness, affecting almost 80% of our sample, along with a presumably high burden of other psychosocial risk factors, somatic illnesses, and genetic predispositions. In fully adjusted models, additionally accounting for mental disorders, non-fatal self-harm remained a strong predictor, indicating elevated mortality risk even among self-harm survivors without psychiatric conditions. These findings highlight the urgency of implementing indicated interventions, such as safety planning [49–51] and suicide-specific brief therapies [52,53], as recommended by WHO [19], for all individuals with a history of self-harm, including those without mental disorders. Immediate post-discharge care is especially critical, given the high risk of reattempt and suicide during this period [13,15].

Our study population included privately insured individuals with higher socioeconomic status and better access to healthcare than the uninsured population relying on South Africa's public sector, who may face greater psychosocial stressors [29]. While self-harm and suicide incidence may be expected to be higher in the public sector, South African evidence suggests that socioeconomic status indicators such as low education are not consistently associated with suicidal behavior [8], supporting the generalizability of our findings. Predictors of self-harm are also generally consistent across public- and private-sector settings. In a study conducted in public primary care clinics in deprived neighborhoods in Cape Town, we found high rates of recurrent suicidal behavior—supporting our finding that non-fatal self-harm is an important risk marker—and strong associations between self-harm, female sex, and mental disorders [54]. Notably, HIV was not associated with self-harm in this study, and similarly, our current analysis found only a weak association, in contrast to findings from high-income countries, where HIV is associated with a 4-fold increase in suicide risk [55].

This study relies on routine data, which has inherent limitations. First, intentional self-harm could be underestimated for several reasons: we could not consider recurrent self-harm episodes; providers might be avoiding self-harm codes to

reduce stigma; patients may underreport self-harm due to fears of stigmatization or involuntary admission [56]; and individuals who did not seek care after self-harm and those who died before accessing care are not captured. Second, mental health and substance use disorder diagnoses were based on claims data and exclude undiagnosed cases. Depression (13%) and anxiety (16%) appeared well ascertained, but substance use disorders were diagnosed in less than 1%—well below expected rates [57,58]. Estimates for substance use should therefore be interpreted with caution, as they likely reflect a small, potentially selected group with more severe, diagnosed conditions. Third, the exact cause of death was not available, preventing an analysis of suicides after non-fatal self-harm. Instead, we examined deaths from unnatural causes, a composite endpoint that includes suicide, accidents, and homicide. While this is a limitation, it also offers advantages: unnatural deaths are more frequent, improving statistical power, and represent a meaningful outcome for prevention. They are also more reliably recorded than specific causes such as suicide [26,59]. Fourth, we combined data from the medical scheme and HIV program to increase sample size, resulting in oversampling of individuals with HIV, which may have introduced bias. Sensitivity analyses restricted to medical scheme data showed no minimal impact on self-harm incidence, its predictors, or its association with subsequent unnatural death, suggesting that oversampling did not materially affect the findings. However, in males, the association between highly lethal methods and unnatural death was not robust in sensitivity analysis, likely due to small numbers in the highly lethal exposure group. We therefore consider this finding exploratory and recommend replication. Nonetheless, results from the primary analysis are clinically plausible and consistent with a Swedish study [31], supporting their credibility. Fifth, the mortality analysis relied on linked vital registration data, and 7% of beneficiaries who could not be linked were excluded. These individuals differed from those included, which may limit generalizability. However, given that they represented a small proportion of the cohort, the impact on findings is likely minimal. Notably, the excluded group had a higher incidence of self-harm and may have been at greater risk of unnatural death, potentially biasing the association towards the null. We therefore consider our estimates conservative.

In conclusion, this study documents a high incidence of self-harm within South Africa's private healthcare sector and provides much-needed local data to support the implementation of context-specific strategies prevention strategies. We identify strong predictors of self-harm that can guide targeted screening strategies, which could be implemented in general medical outpatient settings frequently visited in the weeks preceding self-harm—offering a critical opportunity for early detection and proactive management. Our finding that non-fatal self-harm is a strong risk marker for subsequent unnatural death highlights the need for indicated suicide prevention strategies to prevent recurrent self-harm and suicide in clearly defined high-risk populations.

## Declarations ethical considerations

The Human Research Ethics Committee of the University of Cape Town, South Africa and the Cantonal Ethics Committee, Bern, Switzerland, granted ethical permission for the analysis. Beneficiaries of the medical scheme or their guardians consented to the use of their data in research.

## Declaration of generative AI and AI-assisted technologies in the editing process

During the preparation of this work, we used ChatGPT-4o for proofreading and editing the manuscript to enhance clarity, conciseness, and flow of the text. After using this tool, the authors reviewed and edited the content as needed and take full responsibility for the content of the published article.

## Supporting information

**S1 Strobe Checklist. STROBE, Strengthening the Reporting of Observational Studies in Epidemiology.** https://www.strobe-statement.org/.
(PDF)

**S1 Appendix.  Study protocol.**
(PDF)

**S2 Appendix.  Protocol deviations.**
(PDF)

**S3 Appendix.   Table A.** Incidence rates of self-harm encounters per 100,000 person-years, stratified by sex, for different age groups and overall. **Table B.** Cumulative incidence (%) and 95% confidence intervals of self-harm, stratified by sex, for different age groups and overall. **Table C.** Healthcare utilization before self-harm events compared with matched controls prior to hypothetical matched event dates. **Table D.** Cumulative incidence (%) and 95% confidence intervals of unnatural death, stratified by sex and self-harm method, and overall. **Table E.** Characteristics of the study population by cohort. **Table F.** Characteristics of individuals included and excluded from the mortality analysis. **Fig A.** Incidence of self-harm by age, sex, and HIV status. Incidence per 100,000 person-years in general population as a continuous function of age, stratified by sex and HIV status. Individuals were censored at the time of their first documented self-harm event. Repeated self-harm events were not considered. **Fig B.** Time-varying hazard ratios for unnatural death by sex, comparing individuals with and without prior non-fatal self-harm encounter. Hazard ratios are plotted as a continuous function of analysis time, defined as time since the index self-harm event for individuals with self-harm or baseline for those without self-harm event. Separate models were fitted by sex and adjusted for age. Shaded areas indicate 95% confidence intervals. The dashed horizontal line represents a hazard ratio of 1, indicating no difference in risk. **Fig C.** Incidence of self-harm by age and sex, restricted to medical scheme data. Incidence per 100,000 person-years in general population as a continuous function of age, stratified by sex. Individuals were censored at the time of their first documented self-harm event. Repeated self-harm events were not considered. **Fig D.** Cumulative incidence of self-harm by sex and age group, restricted to medical scheme data. Aalen–Johansen estimate with 95% confidence intervals for cause-specific cumulative over time since baseline or age-group change, stratified by sex and age group. Individuals were censored at the time of their first documented self-harm event. Repeated self-harm events were not considered. **Fig E.** Predictors of self-harm, restricted to medical scheme data. The figure shows age- and sex-adjusted and fully adjusted hazard ratios for predictors of self-harm. Individuals were censored at the time of their first documented self-harm event. Repeated self-harm events were not considered. The error bars represent 95% confidence intervals. Fully adjusted models controlled for age group, sex, HIV, and all mental disorders. * For individuals diagnosed with both depression and bipolar disorder, only the bipolar disorder diagnosis was considered from the date of bipolar diagnosis onwards. Abbreviations: CI, confidence intervals; HR, hazard ratio. **Fig F.** Hazard ratios for unnatural death comparing individuals with and without a history of non-fatal self-harm by sex, restricted to medical scheme data. All models were fitted separately for males and females. Age-adjusted estimates control for age using it as the timeline in Cox model. Fully adjusted models additionally control for HIV status, mental health diagnoses. Error bars represent 95% confidence intervals. Abbreviations: NF-SH, non-fatal self-harm; CI, confidence intervals; HR, hazard ratio. **Fig G.** Hazard ratios for self-harm method as a predictor of unnatural death, by sex, restricted to medical scheme data. All models were fitted separately for males and females. Hazard ratios compare individuals who used highly lethal or less lethal methods at their index self-harm encounter with individuals without a history of non-fatal self-harm (NF-SH). Age-adjusted estimates control for age by using it as the timeline in the Cox model. Fully adjusted models additionally control for HIV status and mental health diagnoses. Error bars represent 95% confidence intervals. Abbreviations: NF-SH, non-fatal self-harm; CI, confidence intervals; HR, hazard ratio.
(DOCX)

## Author contributions

**Conceptualization:** Veronika Whitesell Skrivankova, Stephan Rabie, Mpho Tlali, Naomi Folb, Chido Chinogurei, Sarah Bennett, Aimee Wesso, Yann Ruffieux, Morna Cornell, Soraya Seedat, Matthias Egger, Mary-Ann Davies, Gary Maartens, John Joska, Andreas D. Haas.

**Data curation:** Veronika Whitesell Skrivankova, Chido Chinogurei, Andreas D. Haas.

**Formal analysis:** Veronika Whitesell Skrivankova.

**Funding acquisition:** Matthias Egger, Mary-Ann Davies, Andreas D. Haas.

**Investigation:** Veronika Whitesell Skrivankova, Stephan Rabie, Mpho Tlali, Naomi Folb, Chido Chinogurei, Sarah Bennett, Aimee Wesso, Yann Ruffieux, Morna Cornell, Soraya Seedat, Matthias Egger, Mary-Ann Davies, Gary Maartens, John Joska.

**Methodology:** Veronika Whitesell Skrivankova, Yann Ruffieux, John Joska, Andreas D. Haas.

**Resources:** John Joska.

**Supervision:** Andreas D. Haas.

**Writing – original draft:** Veronika Whitesell Skrivankova, Stephan Rabie, Andreas D. Haas.

**Writing – review & editing:** Mpho Tlali, Naomi Folb, Chido Chinogurei, Sarah Bennett, Aimee Wesso, Yann Ruffieux, Morna Cornell, Soraya Seedat, Matthias Egger, Mary-Ann Davies, Gary Maartens, John Joska.

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
