## [Editor Report · Decision Letter 0]

6 Feb 2025

Dear Dr Haas,

Thank you for submitting your manuscript entitled "Intentional Self-Harm and Unnatural Death in South Africa: A Cohort Study" for consideration by PLOS Medicine.

Your manuscript has now been evaluated by the PLOS Medicine editorial staff as well as by an academic editor with relevant expertise and I am writing to let you know that we would like to send your submission out for external peer review.

Please re-submit your manuscript within two working days, i.e. by Feb 10 2025 11:59PM.

Kind regards,

Suzanne

Suzanne De Bruijn, PhD

Associate Editor

PLOS Medicine

---

## [Decision Letter · Decision Letter 1]

30 May 2025

Dear Dr Haas,

Many thanks for submitting your manuscript "Intentional Self-Harm and Unnatural Death in South Africa: A Cohort Study" (PMEDICINE-D-25-00406R1) to PLOS Medicine. The paper has been reviewed by two subject experts and a statistician; their comments are included below and can also be accessed here: [LINK]

As you will see, the reviewers found the paper interesting and felt that the dataset was very important. However, the raised a large number of questions and felt that further clarification was needed regarding aspects of the methodology and analysis. After discussing the paper with the editorial team and an academic editor with relevant expertise, I'm pleased to invite you to revise the paper in response to the reviewers' comments. Please note that the manuscript will require substantive revisions to address the reviewers’ and editors’ comments, and we plan to send the revised paper to some or all of the original reviewers. At this stage, cannot provide any guarantees regarding publication.

We ask that you submit your revision by June 20th. However, if this deadline is not feasible, please contact me by email, and we can discuss a suitable alternative.

Don't hesitate to contact me directly with any questions (sbruijn@plos.org).

Kind regards,

Heather

Heather Van Epps, PhD

Consulting Editor

[on behalf of]

Suzanne De Bruijn, PhD

Associate Editor

PLOS Medicine

sbruijn@plos.org

Comments from the academic editor:

1. Intentional self-harm is not necessarily suicidal so should not be considered synonymous with a “suicide attempt” (e.g. in the abstract) or suicidal behaviour (background). The authors need to engage with the intentional self-harm literature in the background and explain how it relates to suicidal intentions and attempts in this context. For example, cutting in high-income countries is described as ‘intentional self-harm’ but is conceptualised as a maladaptive coping strategy for emotional regulation and rarely suicidal. In the findings from this paper, cutting seems to be uncommon as a means of intentional self-harm so the authors can make the argument that is more closely linked to suicidal intention.

2. How complete is the coverage of the national vital registration database?

3. It is necessary to present characteristics of the two cohorts separately. The inclusion of a cohort based on HIV status will affect the findings – recommend that the stratified findings are presented in a supplementary file.

4. Please change ‘completed suicide’ to ‘death by suicide’ in line with recommended language to reduce stigma.

Comments from the reviewers:

Reviewer #1 (statistics):

Thanks for the opportunity to read your manuscript. My role is statistical reviewer, so I have focused on the design, data, and analysis that are presented. I have put general comments first, followed by questions relevant to a specific section of the manuscript (with a page/paragraph reference).

This manuscript presents the results of a cohort study investigating incidence and risk factors of self-harm and unnatural deaths in South Africa. Data come from routinely collected sources, a private HIV management program, medical insurance scheme, and a national vital registration database. People aged 10 or over covered the insurance scheme between 2011 and 2020 were included. The main outcomes were the first coded healthcare record for intentional self-harm, and 'unnatural deaths'. A range of covariates including demographics and health status were available. For the mortality data, the primary exposure of interest was a healthcare record for non-fatal self-harm. Healthcare utilisation was compared between exposed and non-exposed, with matching on age and length of follow-up. Overall rates by age and sex were estimated from Poisson regression, and flexible parametric survival models were used for incidence of self-harm (with death as competing risk), including age as covariate and stratified by sex and HIV status. The approach to the analysis is sound (the FSPM is a good choice). I found this to be an interesting and well-written manuscript with well-presented results. I had some requests for clarification about the analysis and data sources, my primary concern was about the interpretation of the multivariable models.

1. It would help in understanding the coverage and limitations if some additional details about the datasets and linkage were provided. In particular, the eligibility and coverage of the medical insurance scheme, HIV disease management program, and vital registration scheme (particularly what information is available from this database). What method of data linkage (deterministic, SLK, probabilistic) was used and on what variables? Why were two databases (insurance + HIV management) used?

2. A limitation of some of the results (e.g. fig 2) with a multivariable model is that the effect estimates have different interpretations depending on the causal model that is assumed ('table 2 fallacy' https://doi.org/10.1093/aje/kws412). Essentially the estimates will be a mix of 'direct-effects' and 'total-effects' which have different interpretations. I would consider either a) proposing a causal model so that the estimates can be interpreted correctly, b) a + concentrating on previous self-harm exposure (in mortality analysis), c )or looking at the data more descriptively.

3. Is it possible that some participants might be recurrent cases, with the data recording intentional self-harm occurring before the study inception date? Effectively, the study has no look-back period, which is concerning as because the length of lookback period affects incidence estimates (e.g. https://doi.org/10.1016/j.jclinepi.2005.12.013)

4. P4, paragraph 3. Can the codes for 'unnatural deaths' be included?

5. P4, paragraph 4. Were the insurance and management program databases the source for these covariates?

6. P4, paragraph 5. Presumably those who died during the first healthcare encounter would have no or very little follow-up data?

7. Was non-fatal self-harm exposure included a fixed covariate, or was this a time-varying covariate?

8. Table S3. With your sample size, even a trivial difference will be 'significant', I would consider putting in an estimate of standardised difference instead of the p-value.

Reviewer #2:

Thank you for the opportunity to review this informative manuscript. The authors present a valuable and rare dataset from South Africa, offering insights into the risk factors for self-harm and its subsequent mortality outcomes in a non-Western setting. While the study could make potentially important contributions to the literature, particularly in under-researched contexts, I have several major concerns related to the methodological clarity, study design, analystical approaches, and conbining two studies in one manuscript.

Comments on the overall study

1. This manuscript presents valuable data with the potential to significantly inform suicide prevention efforts in South Africa and similar contexts. However, the combination of two complex studies, unclear definitions, missing methodological details, and insufficient adjustment for key covariates limit its current clarity and rigor. I would recommend revising the manuscript to either (1) more clearly separate and expand upon the two components, or (2) divide it into two distinct papers with improved methodology and presentation. [Editorial note: we prefer that you do not divide the manuscript into two independent papers]

2. This manuscript combines two conceptually distinct yet related studies: one that investigates risk factors for first-time self-harm in the general population ("self-harm study"), and another that examines the association between self-harm and subsequent unnatural death ("mortality study"). While these topics are connected, they require different study designs, assumptions, and analyses. Presenting both studies in a single paper leads to conceptual and methodological confusion. A clearer distinction in structure and explanation between these two studies is necessary, and it may be more appropriate to report them in separate papers.

3. The study excluded individuals who could not be linked to mortality records. However, no comparison is provided between included and excluded individuals. It is essential to assess whether exclusion may have introduced bias. Basic comparisons of self-harm incidence, sociodemographic composition, and the distribution of mental disorders in the included and excluded samples should be presented. It is unclear whether the exclusion was applied to both the self-harm study and mortality study.

4. In both studies, the use of Cox regression assumes proportional hazards, yet there is no report on whether this assumption was tested. Given that mortality risk after self-harm may decline sharply over time, checking and reporting this assumption is essential.

5. While the authors include some variables such as HIV status and psychiatric diagnoses, important sociodemographic and clinical risk factors are missing. These include physical comorbidities and socioeconomic status, which are known to influence both self-harm risk and mortality.

6. The study treats psychiatric conditions in a relatively broad way. However, certain diagnoses—such as ADHD and conduct disorders—are more relevant in youth, whereas depression and substance use may be important in both younger and older adults. Age-specific modelling of psychiatric risk factors would provide more meaningful insights.

Comments on the self-harm study:

7. The self-harm study attempts to identify first-time self-harm events, but no information is given on how prior episodes were ruled out. A standard approach would involve applying a washout period (e.g., 1-2 years) to reduce the likelihood of misclassifying repeat events as first-time episodes. This methodological limitation should be addressed and discussed.

8. Death was treated as a competing event in the self-harm study where it should have instead been treated as a censoring event.

Comments on the mortality study:

9. The mortality study focuses on unnatural death, a key outcome among people with a history of self-harm. However, natural causes could be also major contributors to subsequent mortality, particularly older self-harm patients. It would strengthen the paper to provide data on all-cause mortality, and then further disaggregate by natural vs. unnatural causes.

10. The authors should report the proportion of deaths with unknown (natural vs unnatural causes) or undetermined (suicide vs accident) causes and discuss their implications.

11. Age is a critical modifier in both the risk of self-harm and its mortality consequences. Yet, the study does not present age interactions or age-stratified analyses. Furthermore, while some subgroup analyses were performed, such as by method lethality and sex, no formal tests of interaction were conducted to assess the statistical evidence of effect modifications.

12. In the mortality study, self-harm is modelled as a fixed exposure. However, self-harm occurs at varying times during follow-up and should be treated as a time-varying covariate in survival analyses. Alternatively, a matched cohort design could strengthen causal inferences. However, the control selection is under-explained. For instance, the index date for controls should correspond to the event date of the matched self-harm case, but this is not clearly stated. Furthermore, no information was given regarding how this matching was implemented. Matching variables included age group (e.g., 40-59 years), but such broad categories can introduce residual confounding. Narrower bands (e.g., 5-year groups) should be considered. The inclusion of sex in the matching process would be also relevant and indicated.

13. The final part of the paper describes post-self-harm healthcare contact within specific time frames (e.g., 1 week, 1 month). This is an important but distinct topic from the core analyses, and would be better suited to a separate, focused manuscript.

Reviewer #3:

Thank you for the opportunity to review this manuscript on a topic of clear importance. The methods are quite sound and the findings are striking and far-reaching, with potential for major impact in the field. I appreciated the extremely sounds methods and statistical approach and concise presentation of a variety of findings, but I believe the manuscript would benefit from some clarifications and added detail in a few areas, outlined below.

1. Abstract: The focus on unnatural death (rather than death by suicide) is logical and the ICD codes used to define unnatural death are appropriate. However, a definition of what constitutes unnatural death would strengthen the abstract.

2. I appreciated the acknowledgement of some of the key limitations in the abstract.

3. Introduction: In paragraphs two and three, "Non-fatal suicidal behavior" seems to be used interchangeably with "suicide attempts", which is not the case. Please address.

4. HIV status is considered in the methods and analytic plan but is completely absent from the abstract and introduction. In the prior literature, there is a link between living with HIV and suicide risk. This should be acknowledged and addressed.

5. Add considerations of how the sample of people who had health insurance coverage might differ from the general population.

6. Considerations of social determinants of health (e.g., food insecurity, housing insecurity, race/ethnicity, socioeconomic status) as predictors of unnatural death in this sample are completely absent. Why weren't these considered, and might it be beneficial to explore them in future studies?

7. It appears the second sub-sample were people living with HIV. Given the prior literature highlighting HIV status as a risk factor for suicide, the implications of using this sub-sample should be addressed more thoroughly. Tables S5 and S6 show higher risk among people with HIV but this is not addressed in the Abstract or Results - please add.

8. Did the prevalence of HIV in the final sample reflect the prevalence in the general population? If people living with HIV were over-sampled, how might this have impacted the generalizability of findings?

9. Similarly, comparison of baseline prevalence of common mental disorders in your sample with the prevalence in the general population will give some indication of generalizability.

---

* Please upload any figures associated with your paper as individual TIF or EPS files with 300dpi resolution at resubmission; please read our figure guidelines for more information on our requirements: http://journals.plos.org/plosmedicine/s/figures. While revising your submission, please upload your figure files to the PACE digital diagnostic tool, https://pacev2.apexcovantage.com/. PACE helps ensure that figures meet PLOS requirements. To use PACE, you must first register as a user. Then, login and navigate to the UPLOAD tab, where you will find detailed instructions on how to use the tool. If you encounter any issues or have any questions when using PACE, please email us at PLOSMedicine@plos.org.

FIGURES AND TABLES

SUPPLEMENTARY MATERIAL

REFERENCES

OBSERVATIONAL STUDIES

* Abstract: Please include the study design, population and setting, number of participants, years during which the study took place (enrollment and follow up), length of follow up, and main outcome measures.

* Please ensure that the study is reported according to the STROBE (or appropriate STOBE extension) guideline (available from: https://www.equator-network.org/reporting-guidelines/strobe) and include the completed STROBE (or STROBE extension) checklist as Supporting Information. Please add the following statement, or similar, to the Methods: "This study is reported as per the Strengthening the Reporting of Observational Studies in Epidemiology (STROBE) guideline (S1 Checklist)." When completing the checklist, please use section and paragraph numbers, rather than page numbers.

* For all observational studies, in the manuscript text, please indicate: (1) the specific hypotheses you intended to test, (2) the analytical methods by which you planned to test them, (3) the analyses you actually performed, and (4) when reported analyses differ from those that were planned, transparent explanations for differences that affect the reliability of the study's results. If a reported analysis was performed based on an interesting but unanticipated pattern in the data, please be clear that the analysis was data driven.

* Please state in the Methods section whether the study had a prospective protocol or analysis plan. If a prospective analysis plan (from your funding proposal, IRB or other ethics committee submission, study protocol, or other planning document written before analyzing the data) was used in designing the study, please include the relevant document(s) with your revised manuscript as a Supporting Information file to be published alongside your study and cite it in the Methods section. A legend for this file should be included at the end of your manuscript. If no such document exists, please make sure that the Methods section transparently describes when analyses were planned, and when/why any data-driven changes to analyses took place. Changes in the analysis, including those made in response to peer review comments, should be identified as such in the Methods section of the paper, with rationale.

---

## [Decision Letter · Decision Letter 2]

30 Jul 2025

Dear Dr Haas,

Many thanks for submitting your manuscript "Self-Harm and Unnatural Death in South Africa: A Cohort Study" (PMEDICINE-D-25-00406R2) to PLOS Medicine. The paper has been re-reviewed by the subject experts and a statistician; their comments are included below and can also be accessed here: [LINK]

As you will see, the reviewers think the manuscript has improved, but have some remaining concerns. After discussing the paper with the editorial team and an academic editor with relevant expertise, I'm pleased to invite you to revise the paper in response to these last reviewers' comments. Specifically, R#1 comments on the revision being positioned as prognostic, whereas the results are still appropriate for a causality study. R#2 would like you to test for sex-differences, and if these are not present this reviewer suggests to combine the estimates. Please note that we do not expect you to address the comments regarding reporting all-cause mortality numbers as well. We plan to send the revised paper to some or all of the original reviewers, and we cannot provide any guarantees at this stage regarding publication.

We ask that you submit your revision by Aug 20 2025 11:59PM. However, if this deadline is not feasible, please contact me by email, and we can discuss a suitable alternative.

Don't hesitate to contact me directly with any questions (sbruijn@plos.org).

Best regards,

Suzanne

Suzanne De Bruijn, PhD

Associate Editor

PLOS Medicine

sbruijn@plos.org

Comments from the academic editor:

The recommendations regarding the sex differences are valid, and should be addressed.

The suggestion of reporting all-cause mortality separately to unnatural deaths is beyond the scope of this paper.

Comments from the reviewers:

Reviewer #1: Thanks for the revised manuscript and responses to my original review. The updates have clarified most of my original queries.

My remaining concern is that the updates to the manuscript now position the aim of the study to be prognostic, rather than to estimate a causal relationship between the different study covariates and risk of self-harm and unnatural deaths. However, the methods described and the way the results are reported are for an aetiological study rather than a prediction study. If the aim was a predictive model of the study outcomes, the study needs a) a measure of discrimination of the model (e.g. c-statistic), b) level of calibration of predictions from the model, c) a way of measuring or managing model optimism (cross-validation, bootstrapping), and d) a clear vision of the 'use case' of the model, i.e. where it could be implemented and what actions could be taken with predictions from the model are available. The individual coefficients from the model should not be interpreted as effect estimates either (e.g. see Box 2 of van Diepen 2017 which was referenced in the response). The manuscript should report appropriate results of whatever approach is chosen as the aim of the study (descriptive, predictive, causal, see this recent reference summarises this well: https://doi.org/10.1002/sim.10244)

Reviewer #2: I thank the authors for their responses. I have two further comments on the revised paper as outlined below:

1. The authors argued that "The aim of the mortality analysis is to inform suicide prevention efforts. Natural causes of death are not preventable through suicide prevention interventions, and elevated risk may reflect higher comorbidity in this population. To maintain conceptual clarity and focus, we therefore prefer to limit our analysis to unnatural deaths." However, suicide prevention activities that incorporate psychosocial support may have spillover effects on overall health, including physical health. Moreover, a holistic approach to the care of individuals who have self-harmed should address both mental and physical health needs. Importantly, all-cause mortality is a key overarching outcome for this population, and estimating its risk can help contextualise the relative contributions of natural and unnatural causes of death. Therefore, I wonder if the authors could consider providing risk estimates for all-cause and natural-cause mortality, at least in the supplementary materials, if not in the main text, to offer a more comprehensive understanding of mortality risk in this population.

2. In Figure 4, sex-specific analyses were presented. However, the 95% confidence intervals for hazard ratios in males and females overlap substantially, suggesting no statistically meaningful sex difference. It would be worthwhile to formally test for sex differences by including a sex-by-exposure interaction term in the regression models. If no statistical evidence for interaction is found, presenting combined estimates for males and females may be more appropriate.

Reviewer #3: I thank the authors for their careful and thorough attention to the reviewer comments. My comments have now been adequately addressed.

---

* Financial disclosure: Thank you for including your financial enclosure. Please also include URLs from the funders in this statement.

* Ethics statement: Could you please include IRB approval numbers?

* Thank you for including an Author summary.

-Can you please limit the number of bullet points under 'What Did The Researchers Do And Find' to 3?

-Can you please include limitations in the final bullet point of 'What Do These Findings Mean?

---

## [Decision Letter · Decision Letter 3]

9 Sep 2025

Dear Dr. Haas,

Thank you very much for re-submitting your manuscript "Self-Harm and Unnatural Death in South Africa: A Cohort Study" (PMEDICINE-D-25-00406R3) for review by PLOS Medicine.

I have discussed the paper with my colleagues and the academic editor and it was also seen again by 2 reviewers. I am pleased to say that provided the remaining editorial and production issues are dealt with we are planning to accept the paper for publication in the journal.

We look forward to receiving the revised manuscript by Sep 16 2025 11:59PM.   

Sincerely,

Suzanne De Bruijn, PhD

Associate Editor 

PLOS Medicine

plosmedicine.org

Requests from Editors:

GENERAL EDITORIAL REQUESTS

* Please change your title to: "Self-harm incidence rates and their predictive value for subsequent unnatural death in South Africa: a Cohort Study"

* Please ensure that all abbreviations are defined at first use throughout the text.

* Please confirm that all numbers presented in the abstract are present and identical to numbers presented in the main manuscript text.

GENERAL

* Statistical reporting: Please revise throughout the manuscript, including tables and figures.

- Please report statistical information as follows to improve clarity for the reader ""22% (95% CI [13,28]; p</=)"".

- Please separate upper and lower bounds with commas instead of hyphens as the latter can be confused with reporting of negative values.

- Please repeat statistical definitions (HR, CI etc.) for each set of parentheses."

* Please consider moving some of the methods from the supplements into the main text.

*ABSTRACT: please mention clearly that 2011-2022 were the years for which you have data.

FUNDING STATEMENT

* Thank you for including your funding statement. Could you please include the initials of the people who obtained the grant?

DATA AVAILABILITY

* Thank you for providing a github/Zenodo link in the data availability statement in the manuscript. Can you please ensure the same statement, including the code availability, is also present in the metadata.

FIGURES

* Please consider if moving some of the figures shared in the supplementary information into the main text would aid the reader. Specifically, we would suggest to move Figure A from the supplements to the main text.

OBSERVATIONAL, COHORT, CROSS-SECTIONAL, AND CASE CONTROL STUDIES

*Thank you for including the STROBE checklist, and mentioning this in your methods section. Could you please spell out STROBE? "This study is reported as per the Strengthening the Reporting of Observational Studies in Epidemiology (STROBE) guideline (Appendix S4).""

-When completing the checklist, please use section and paragraph numbers, rather than page numbers.

Comments from Reviewers:

Reviewer #1: Thanks for the revised manuscript and responses to my original review. The explanation given in the responses and the corresponding updates was helpful for me. The updates to the wording to specifically position the manuscript as a 'prognostic factor' study (along with some adjustment of terminology) works well.

Reviewer #2: I have no further comments.

---

## [Editor Report · Decision Letter 4]

17 Sep 2025

Dear Dr Haas, 

On behalf of my colleagues and the Academic Editor, Charlotte Hanlon, I am pleased to inform you that we have agreed to publish your manuscript "Incidence and Prognostic Factors of Self-Harm and Subsequent Unnatural Death in South Africa: A Cohort Study" (PMEDICINE-D-25-00406R4) in PLOS Medicine.

PRESS

Sincerely, 

Suzanne De Bruijn, PhD 

Associate Editor 

PLOS Medicine